# CONVINT: A SEMI-STRUCTURED INTENTION FRAMEWORK FOR CONVERSATIONAL UNDERSTANDING

## ABSTRACT

Understanding user intentions is critical for conversational AI, especially with the rise of large language models (LLMs) that demand a more nuanced comprehension of dialogue. Existing approaches, relying on rigid slot-value structures or unstructured representations, often miss the complexity of human intentions. In this work, we propose ConvINT, a novel semi-structured intention framework that offers a more holistic and fine-grained understanding of user intentions by organizing them into four key aspects: situation, emotion, action, and knowledge. Grounded in psychological and cognitive intention theories, ConvINT provides LLMs with a richer context for understanding user inputs while offering a semi-structured format that seamlessly integrates with prompt-based intention learning. To enable the efficient adoption of this framework, we introduce a Weakly-supervised Reinforced Generation (WeRG) method that scales ConvINT annotations across large datasets with high quality. By combining a small set of human-annotated instances with coarsely labeled data as weak supervision signals, WeRG effectively learns to generate ConvINT annotations, ensuring both scalability and precision. Experimental results demonstrate that integrating ConvINT with WeRG markedly improves LLMs' ability to comprehend user intentions, yielding significant gains in downstream tasks such as response generation and task completion, as validated by both automatic metrics and human evaluations. These findings highlight ConvINT's potential as a comprehensive and adaptable framework for advancing intention understanding in conversational AI.

## 1 INTRODUCTION

Recent advancements in conversational systems designed for social support and functional services—such as conversational recommendation (Li et al., 2018; Kang et al., 2019) and emotional support (Liu et al., 2021a; Zheng et al., 2023)—have garnered growing interest from both academia and industry. A key upstream component in these systems is Conversational Understanding (CU), which focuses on accurately interpreting user inputs from multiple perspectives (Qin et al., 2020; Park et al., 2021; Chen et al., 2022b; Wang et al., 2023c). This component plays a fundamental role in driving downstream tasks, such as policy planning (Kwan et al., 2023) and response generation (Hosseini-Asl et al., 2020; Wang et al., 2023a), by providing structured and interpretable representations of user intentions.

Typically, CU parses user intentions into structured semantic representations based on predefined conversational ontologies, which include specified intent classes and structured slot-value pairs (Casanueva et al., 2020; Tang et al., 2023; Pham & Nguyen, 2024). While these methods have been effective in constrained scenarios, they face significant limitations in real-world applications due to their reliance on structured ontologies, making it challenging to accommodate evolving user needs and complex conversational nuances (Zhang et al., 2021b; De Raedt et al., 2023; Nguyen et al., 2023; Liang et al., 2024b;a). Moreover, many existing methods focus on single-turn intent detection and slot labeling, often resulting in shallow and fragmented interpretations that fail to capture the fluid dynamics of multi-turn dialogues (Zhang et al., 2022; Zhou et al., 2023). This rigidity hampers the system's ability to understand the deeper layers of user intent, which often encompass emotions, evolving contexts, and knowledge states (Wang et al., 2023a).

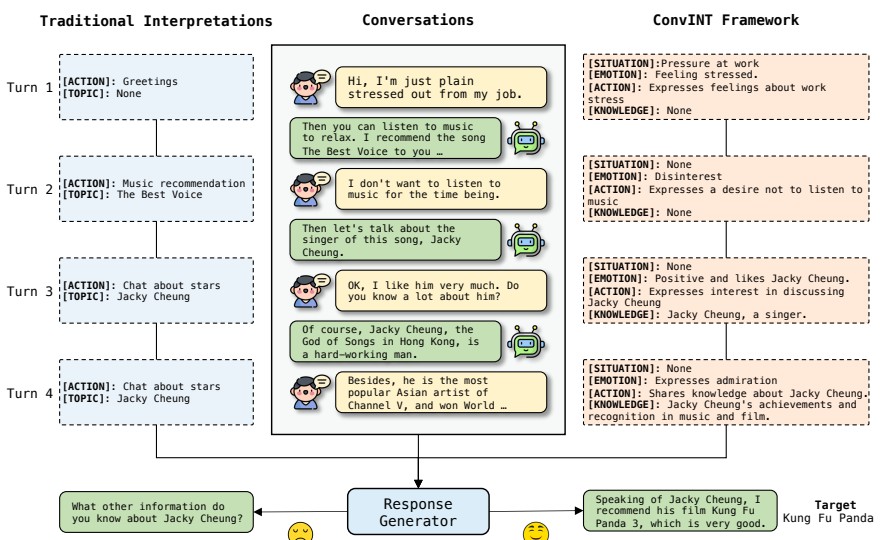

Figure 1: A comparison of existing structured interpretations and the proposed ConvINT framework.

Recently, Large Language Models (LLMs) (Ouyang et al., 2022; Chiang et al., 2023; Jiang et al., 2023; OpenAI, 2023; Dubey et al., 2024) have revolutionized conversational AI with their exceptional capabilities in context understanding and generalization. However, current CU interpretations remain oversimplified and lack the flexibility needed to fully exploit the capabilities of LLMs, thereby preventing them from comprehending the richness and depth inherent in real-world conversations. This gap becomes even more pronounced when conversational systems need to handle intricate aspects such as user emotions, situational contexts, intended actions, and evolving knowledge—key elements that structured representations fail to capture. In an effort to overcome these limitations, an alternative approach involves summarizing conversation histories into free-text descriptions, allowing for a more flexible and comprehensive capture of conversational details without the constraints of fixed structures (Liu et al., 2019; Wu et al., 2021; Chen et al., 2021; 2022a; Yang & Zhu, 2023). While this approach retains more information, it often becomes unfocused, prone to inconsistencies, and tends to overlook the core elements of user intentions. Furthermore, the unstructured nature of free-text outputs makes it challenging to train and evaluate CU models effectively, ultimately limiting their practical applicability in capturing the richness and depth needed for accurate intention understanding.

To address these challenges, we introduce ConvINT, a novel semi-structured intention framework that provides a more comprehensive, aspect-aware, and flexible approach to effective CU. As shown in Figure 1, inspired by psychological and cognitive intention theories (Schröder et al., 2014), ConvINT organizes user intentions into four fundamental aspects: (1) situation, which captures the conversational context; (2) emotion, reflecting the user's psychological state; (3) action, representing the intended actions; and (4) contextual knowledge, encompassing the evolving information throughout the dialogue. Compared with existing CU interpretations, *e.g.*, rigidly parsing user intentions into elements like *chat about stars* and *Jacky Cheung* within strictly structured ontologies, this structured yet flexible organization enables LLMs to access a richer, more nuanced understanding of user intentions, making ConvINT particularly well-suited for integration with prompt-based intention learning.

To facilitate the large-scale application of this framework, we develop a Weakly-supervised Reinforced Generation (WeRG) approach to efficiently expand ConvINT annotations across extensive datasets. Specifically, WeRG first constructs a set of supervised fine-tuning data from diverse sources with coarse-to-fine labels, including a large proportion of existing intents and LLM-annotated supervisions, as well as a limited set of human-annotated ConvINT data. Recognizing that existing intents and LLM-generated annotations can be noisy, and that human annotations are scarce, WeRG synergistically combines these diverse annotations as weak supervision signals, assigning varying rewards to each data source conditioned on their coarse-to-fine levels. By fine-tuning the model with reinforcement learning, WeRG effectively facilitates the training of a conditional policy model that maximizes the utility of human-crafted annotations while efficiently generating high-quality Con-

vINT data. We thoroughly evaluate the quality and effectiveness of the newly generated ConvINT data, revealing not only the superiority of the WeRG method but also ConvINT's capability to elevate the performance of downstream response generation tasks. This underscores feasible future directions for large-scale dataset construction and model training in CU scenarios.

To sum up, our contributions are as follows:

- We draw from interdisciplinary intention theories to formulate the Conversational INTention (ConvINT) framework, a fine-grained, aspect-aware method effective in facilitating an in-depth understanding of intricate conversational intentions.
- We devise an efficient Weakly-supervised Reinforced Generation (WeRG) mechanism that synergizes various sources of annotated data for the model fine-tuning, thereby achieving high-quality ConvINT data acquisition.
- Utilizing the WeRG method, we first construct a high-quality ConvINT dataset for conversational understanding. In-depth analysis further demonstrates that the generated ConvINT data can significantly enhance downstream conversational tasks.

## 2 RELATED WORKS

**Conversational Understanding.** CU is an essential, yet challenging research topic in conversational AI (Zhang & Zhao, 2021; Chen et al., 2022b; Liu et al., 2023). Its primary goal is to summarize user inputs at each turn throughout conversations into precise semantic interpretations. To achieve this, early efforts relied on static and structured conversational ontologies, delving into the individual tasks of intent detection and slot filling. These approaches primarily developed separate models for categorizing intents and marking slots, in which significant progress has been made. (Yao et al., 2014; Ravuri & Stolcke, 2015; Vu et al., 2016; Kurata et al., 2016; Xia et al., 2018; Lee & Jha, 2019; Casanueva et al., 2020; Tang et al., 2023; Zhang et al., 2023; Li et al., 2023; Mullick et al., 2024). Considering the close correlation between these tasks, recent efforts have shifted focus to investigating joint intent-slot recognition that leverages a joint model to simultaneously predict intents and slot sequences (Zhang et al., 2019; Qin et al., 2021b; Weld et al., 2023; Mirza et al., 2024; Yin et al., 2024; Pham & Nguyen, 2024). For example, certain approaches facilitating simultaneous intent detection and slot filling leverage shared parameters (Liu & Lane, 2016; Wang et al., 2018), while others learn the relationship between the two via various interaction flows (Goo et al., 2018; Qin et al., 2019; 2021a). While these methods have shown progress, their reliance on static conversational ontologies limits their applicability in real-world scenarios, where unforeseen user needs continually evolve.

Motivated by this challenge, research in this field also explores discovering new intents, slots, and values beyond the scope of static and structured ontologies. Innovations have developed techniques like unsupervised learning methods (Xie et al., 2016; Yang et al., 2017; Caron et al., 2018; Zhang et al., 2021a; Yu et al., 2022; De Raedt et al., 2023; Nguyen et al., 2023) and semi-supervised learning methods (Hsu et al., 2018; 2019; Zhang et al., 2021b; 2022; Zhou et al., 2023; Liang & Liao, 2023; Liang et al., 2024b;a; Wu et al., 2024). Extending beyond the inherently structured nature of the above semantic interpretations, alternative methods (Liu et al., 2019; Wu et al., 2021; Chen et al., 2021; 2022a; Yang & Zhu, 2023) propose summarizing conversation content into concise, free-form text descriptions, facilitating more effective CU by providing greater flexibility in capturing conversational nuances without the constraints of rigid ontologies. Yet, the challenge persists in the lack of an effective framework capable of balancing the grasp of in-depth information in conversations while guiding the focus on producing accurate semantic interpretations, a gap that this work addresses by introducing the semi-structured ConvINT framework into CU.

**Fine-tuning Techniques for LLMs.** In recent years, LLMs have witnessed substantial advancements, showcasing remarkable capabilities in natural language understanding and generation. By fine-tuning with specific application data, these large-scale models can be further adapted for downstream use cases. Generally, the fine-tuning of LLMs has mainly been approached in two ways. The first line of methods focuses on Supervised Fine-Tuning (SFT) (Ding et al., 2023; Xu et al., 2024), which updates the LLMs' parameters directly using well-crafted SFT data through supervised learning objectives, such as maximum likelihood estimation. Along this line, some studies (Chiang et al., 2023; Geng et al., 2023; Xu et al., 2024) have delved into designing high-quality data to facilitate the

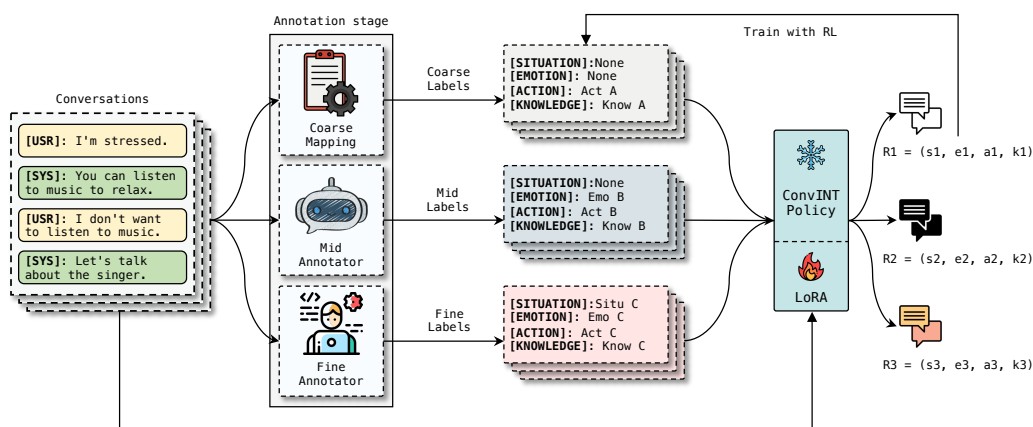

Figure 2: An overview of the proposed ConvINT framework and WeRG mechanism.

SFT process. Additionally, recent efforts also explore developing Parameter-Efficient Fine-Tuning (PEFT) methods to balance the quality and efficiency in the SFT process (Lester et al., 2021; Hu et al., 2022; Zaken et al., 2022; Zhang et al., 2024). Another branch of fine-tuning approaches is Reinforcement Learning Fine-tuning (RLFT). In RLFT, a reward model is developed using feedback directly derived from human preferences, which is then employed to fine-tune LLMs through an RL objective to maximize the reward (Jaques et al., 2019; Ouyang et al., 2022; Korbak et al., 2022; Rafailov et al., 2023; Wu et al., 2023; Wang et al., 2024). As LLMs continue to evolve to be capable of supervising other models, a new method named RL from AI Feedback (RLAIF) has gained popularity (Bai et al., 2022). RLAIF utilizes the natural language feedback generated by the LLMs to self-improve task instructions, thereby optimizing LLMs to be harmless and detoxified (Shinn et al., 2023; Madaan et al., 2023; Hao et al., 2023).

In both SFT and RLFT methods, collecting high-quality supervision or reward signals is vital for enhancing LLMs' fine-tuning performance. Yet, this process can be financially costly and often yields a significant amount of substandard data, leading to compromised fine-tuning outcomes. This work addresses these issues with the WeRG method, which synergistically combines coarse to fine-level data as weak supervision signals to facilitate the RL process without incurring additional costs.

## 3 METHOD

In this section, we detail our approach to effective conversational understanding with the framework illustrated in Figure 2. We first introduce the preliminaries of conversational understanding scenarios in Section 3.1. Subsequently, we formulate the ConvINT framework for grasping fine-grained, aspect-aware information throughout the conversational process (Section 3.2) and introduce the WeRG mechanism for synergistically combining various coarse to fine data sources to efficiently generate ConvINT data (Section 3.3).

### 3.1 PRELIMINARIES

In this work, we study the task of conversational understanding as follows: Consider a conversation dataset represented as $\mathcal{D} = \{h_i, x_i, y_i\}_{i=1}^{N}$, where $N$ is the total number of training instances. Suppose $x_i$ denotes a user utterance $u_t$ at the $t$-th turn of a conversation, and $y_i$ is its corresponding system response. In this context, $h_i$ refers to the historical information preceding $x_i$, e.g., $h_i = \{u_1, y_1, \ldots, u_{t-1}, y_{t-1}\}$. The primary objective is to learn a model, $\mathcal{M}$, to generate a collection of ConvINT data, $\mathcal{O} = \{\langle s_i, e_i, a_i, k_i \rangle\}_{i=1}^{N}$, based on each user utterance $x_i$ and its corresponding historical context $h_i$, using a set of weak supervision signals from various data sources:

$$f_{\mathcal{M}} : (h_i, x_i) \rightarrow o_i, \tag{1}$$

where $o_i = \langle s_i, e_i, a_i, k_i \rangle \in \mathcal{O}$ corresponds to the spans of situation, emotion, action, and knowledge, respectively. As such, we can evaluate the effectiveness of these generated ConvINT data in interpreting intricate conversations and further utilize them to enhance downstream tasks.

## 3.2 ConvINT Framework

Here, we detail the formulation of the proposed ConvINT framework for conversational understanding. Given a user utterance and its dialogue history, many existing CU methods primarily focus on interpreting these them using simplistic and structured elements like intents and slot-value pairs. However, these rigid semantic interpretations often fail to capture the rich and in-depth user information inherently conveyed within the conversational context, including conversation features, user emotional status, behavioral characteristics, and contextual knowledge. Motivated by this, we formulate the ConvINT frame, a formalism that reflects the above aspects and establishes a fine-grained, multidimensional structure based on the comprehensive analysis of conversational dynamics.

Specifically, ConvINT draws inspiration from semantic pointers (Eliasmith, 2013; Blouw et al., 2016) to grasp the nature of intentions and their breakdown within conversational scenarios. According to the intention theories in psychological and cognitive sciences (Schröder et al., 2014), we formulate the fine-grained and aspect-aware ConvINT framework as follows:

**Definition:** *Conversational intentions are semantic pointers that bind together information about situations, emotional evaluations, actions, and sometimes also about self-knowledge.*

Building upon this formalism, we further elaborate on the concepts of situation, emotion, action, and knowledge as below:

*[SITUATION]*: Describe physical or situational features of the current conversation.

*[EMOTION]*: Capture any emotional states or evaluations expressed by the user.

*[ACTION]*: Refer to any actions the user mentions taking to achieve within their utterances.

*[KNOWLEDGE]*: Identify entities and relevant knowledge mentioned in the context.

With this design, we can break down user inputs into four key aspects, gaining deeper insights into the intentions behind their utterances. Additionally, each aspect of the ConvINT frame can be expressed in free-form natural language rather than being limited to a predefined and static conversational ontology, offering greater flexibility in accurately understanding users' evolving needs.

## 3.3 WeRG mechanism

After formulating the ConvINT framework that is capable of capturing enriched and in-depth information to understand complex conversations, we need to acquire annotated ConvINT data for evaluation and further downstream applications. To accomplish this, a straightforward method involves directly annotating high-quality ConvINT data using human annotators and performing SFT to optimize LLMs for generation. Despite its effectiveness, this method is labor-intensive and financially costly. Alternatives include leveraging cost-effective LLMs as annotators or directly transforming existing simplistic semantic interpretations, such as intents and slot-value pairs, into ConvINT labels for supervising LLMs. However, the resulting annotations are prone to noise, failing to cover the fine-grained aspects present in the ConvINT frame, which leads to degraded performance.

Given the above considerations, we thereby devise an effective weakly-supervised reinforced generation mechanism. Intuitively, WeRG is designed to synergistically integrate various sources of annotations with coarse-to-fine labels as weak supervision signals, thereby ensuring both efficiency and high quality in the generation of ConvINT data. To achieve this, consider a conversation dataset $\mathcal{D} = \{h_i, x_i, y_i\}_{i=1}^{N}$, we first collect a WeRG fine-tuning dataset, $\mathcal{D}_{\text{WeRG}} = \mathcal{D}_{\text{coarse}} \cup \mathcal{D}_{\text{mid}} \cup \mathcal{D}_{\text{fine}}$, by employing a variety of annotation methods. Specifically, $\mathcal{D}_{\text{coarse}}$ utilizes hard mapping to transform existing structured interpretations into ConvINT labels, yielding coarse-level labels. In contrast, $\mathcal{D}_{\text{mid}}$ prompts cost-effective LLMs to annotate conversations within the ConvINT frame. Since LLMs can extract more nuanced information than existing structured interpretations, $\mathcal{D}_{\text{mid}}$ is thus endowed with mid-level labels. Unlike the above, $\mathcal{D}_{\text{fine}}$ employs human annotators to create ConvINT data, thereby providing high-quality fine-level labels. Notably, due to the high cost of human annotation, the number of examples in $\mathcal{D}_{\text{fine}}$ is significantly less than those in $\mathcal{D}_{\text{mid}}$ and $\mathcal{D}_{\text{coarse}}$. Here, as our primary focus is on the generation of ConvINT data, we formally redefine the conversation dataset as follows: $\mathcal{D}_{\text{WeRG}} = \{(h_i, x_i, o_i)\}_{i=1}^{|\mathcal{D}_{\text{WeRG}}|}$.

To effectively utilize the coarse-to-fine level signals within $\mathcal{D}_{\text{WeRG}}$, following Wang et al. (2024), we further enhance $\mathcal{D}_{\text{WeRG}}$ by incorporating weak and tiered reward signals, which are meticulously calibrated to account for the variations across different annotation methods. Specifically, the reward is structured as a quadruple as follows:

$$r_c(h_i, x_i, o_i) = \langle r_s^{c_i}, r_e^{c_i}, r_a^{c_i}, r_k^{c_i} \rangle, \text{where } c_i \in \{\text{coarse}, \text{mid}, \text{fine}\}, \tag{2}$$

where $\langle r_s^{c_i}, r_e^{c_i}, r_a^{c_i}, r_k^{c_i} \rangle$ are simple scalar rewards aligning with the $\langle s_i, e_i, a_i, k_i \rangle$ aspects in $o_i$. Notably, unlike previous studies such as those described by Wang et al. (2024) that regard the entire ground-truth sequence equally, this quadruple reward allows for the allocation of distinct reward components to each aspect of the ConvINT labels based on the level of information provided by the annotations. Meanwhile, by establishing the reward hierarchy as $r_{\text{coarse}} < r_{\text{mid}} < r_{\text{fine}}$, we can effectively guide the fine-tuning of LLMs towards favoring higher-quality ConvINT data.

Given the constructed fine-tuning dataset $\mathcal{D}_{\text{WeRG}}$ and the reward information $r_c(h, x, o)$, we thereafter optimize a KL-regularized RL objective to fine-tune an LLM policy $\pi_\theta$ for efficiently generating high-quality ConvINT data as follows:

$$J_{\text{WeRG}}(\theta) = \mathbb{E}_{\mathcal{O} \sim \pi_\theta}[r_c(h, x, o)] - \beta D_{KL}(\pi_\theta, \pi_w), \tag{3}$$

where $\pi_w$ denotes the policy model augmented by the weak supervision signals in $\mathcal{D}_{\text{WeRG}}$. As demonstrated by previous works (Peters & Schaal, 2007; Korbak et al., 2022; Rafailov et al., 2023; Wang et al., 2024), the optimal solution $\pi^*$ for the Equation (3) can be described as follows:

$$\pi^*(o|h, x, c) = \arg \max_\theta J_{\text{WeRG}}(\theta) \propto \pi_w(o|h, x, c) \exp \left( \frac{1}{\beta} r_c(h, x, o) \right). \tag{4}$$

Based on this optimal solution, the KL-regularised RL objective can be cast as minimizing the KL divergence of policy $\pi_\theta$ from the this policy $\pi^*$ under the WeRG fine-tuning dataset $\mathcal{D}_{\text{WeRG}}$ (Nair et al., 2020; Korbak et al., 2022; Wang et al., 2024):

$$\pi_\theta = \arg \min_\theta \mathbb{E}_{(h,x,c) \sim \mathcal{D}_{\text{WeRG}}} \left[ D_{KL} \left( \pi^*(\cdot|h, x, c) \parallel \pi_\theta(\cdot|h, x, c) \right) \right]. \tag{5}$$

With this WeRG approach, we can effectively utilize weak supervision signals gathered from diverse data sources with coarse-to-fine labels, thereby enabling the LLM policy model to optimally generate ConvINT data.

## 4 EXPERIMENTS

### 4.1 DATASETS

We conduct experiments on two conversational datasets—**DuRecDial** (Liu et al., 2021b) (recommendation dialogues) and **ESConv** (Liu et al., 2021a) (emotional support dialogues)—to evaluate the proposed ConvINT framework and WeRG mechanism. Specifically, **DuRecDial** is a dataset of conversational recommendations that consists of 16.5K English-Chinese parallel dialogues and approximately 255K natural language utterances, along with 14 goals and 646 topics. We utilize the English version of the dataset for our experiments. **ESConv** is an emotional support conversation dataset comprising 1,300 cases with 8 distinct support strategies. Each case is accomplished by a specified problem type, an emotion type, and a detailed situation description. For both datasets, we maintain the same training, development, and test splits as previous studies (Dao et al., 2023; Deng et al., 2024; He et al., 2024). More experimental details are provided in the Appendix A.

### 4.2 EVALUATION PROTOCOLS

In this work, the primary goal is to evaluate the quality of the ConvINT data generated via the WeRG approach, specifically its capability to capture the fine-grained aspect information as formulated by the ConvINT framework. To achieve this, we engage human annotators to label the ConvINT labels for the test set, thereby establishing the fundamental ground truth for the quality evaluation. After acquiring the ConvINT data, we also aim to validate its functionality in downstream applications. To this end, we further apply the generated ConvINT data to target-driven conversation scenarios, evaluating its effectiveness in enhancing the ability of conversational agents to respond to users and

guide them toward the ultimate targets. In light of the above considerations, the evaluation protocols used in our experiments can be broadly categorized as follows:

**Automatic Evaluation Protocols.** The acquisition of ConvINT data via the WeRG method is fundamentally a generative process. In this sense, with the ground-truth labels previously established, most existing automatic generation metrics can be applied to assess the quality of the generated ConvINT data. Specifically, we utilize word-level F1 (**F1**) and **BLEU-N** (N=1, 2) metrics (Papineni et al., 2002) to compute the lexical overlap between the generated ConvINT data and the ground-truth labels, offering a quantitative measure of the precision and syntactic accuracy of the WeRG method. Additionally, we adopt **BERTScore** (Zhang et al., 2020) and **BARTScore** (Yuan et al., 2021) to measure the semantic similarity, further evaluating how well the generated data contextually aligns with the ground truth. For validating the effectiveness of the ConvINT data in downstream tasks, we measure the dialogue-level Success Rate (**SR**) and the Averaged number of conversation Turns (**Avg. Truns**) necessitated to successfully guide users to targets (Lei et al., 2020a;b).

**Human-centered Evaluation Protocols.** Generally, the most effective method for evaluating such texts is still human evaluation, wherein human annotators assess the quality of the generated ConvINT data. This evaluation can be approached from various perspectives, and we suggest several commonly used methodologies (Zheng et al., 2024): (1) **Informativeness (Info.)**: can the ConvINT data capture the key information throughout the conversation process? (2) **Understanding (Und.)**: whether the ConvINT data is clear and easy to understand in accurately describing users' real intentions? (3) **Conciseness (Con.)**: does the ConvINT data effectively communicate the necessary details without superfluous content? For these evaluations, we engaged three students as annotators, each tasked with assessing the ConvINT labels generated by various methods in 50 randomly selected conversations to ensure a comprehensive comparison.

### 4.3 BASELINES

In our experiments, we explore prompting LLMs for two different ways of generating ConvINT data as the baselines ((Appendix B).

**Direct Prompt** (Brown et al., 2020). Directly provide LLMs with the necessary instructions as prompts to generate ConvINT data that grasp user intentions throughout the conversation process, including zero-shot and few-shot settings. In particular, the few-shot demonstrations are randomly selected from a set of manually constructed ConvINT examples.

**Chain-of-Thought (CoT) Prompt.** Building upon manually created examples provided, equip LLMs with detailed task descriptions and explanations of the ConvINT framework, specifying the criteria for generating ConvINT data by referring to the CoT method (Yao et al., 2023; Wang et al., 2023b), also including zero-shot and few-shot settings similar to the Direct Prompt baseline.

### 4.4 MAIN RESULTS

#### 4.4.1 AUTOMATIC EVALUATION RESULTS

To demonstrate the quality of the ConvINT data generated via the proposed WeRG mechanism, we compare our method against other baselines, with results reported in Table 1.

Firstly, regarding the content-based evaluation metrics, such as F1 and BLEU-1/2, our method consistently surpasses all baselines by a noticeable margin on both datasets. Among them, the zero-shot CoT Prompt demonstrates superior performance over the zero-shot Direct Prompt by enriching LLMs' prompts with more detailed task descriptions and ConvINT explanations. The few-shot CoT Prompt further amplifies this superiority by incorporating manually crafted ConvINT examples, showcasing the advantages of high-quality data in facilitating ConvINT data generation. Notably, our method synergistically integrates various sources of data annotated with coarse-to-fine labels to perform RLFT, allowing for a more effective and robust ConvINT model.

Secondly, in terms of similarity-based evaluation metrics such as BERTScore and BARTScore, our method excels in generating more detailed and comprehensive content with a broader inclusion of key information that semantically aligns with each aspect defined in the ConvINT frame. The baseline methods, without explicit guidance to favor more high-quality ConvINT data, are limited in yielding outcomes that adequately reflect the depth and richness required by the ConvINT frame.

Table 1: Automatic evaluation of ConvINT generation performance. Results in bold indicate significant superiority over other methods. Direct and CoT Prompt represent zero-shot baselines, while -*w/* example indicates the few-shot baseline setting.

| Methods | F1 ↑ | BLEU1 ↑ | BLEU2 ↑ | BERTScore ↑ | BARTScore ↓ |
|---|---|---|---|---|---|
| *DuRecDial* | | | | | |
| Direct Prompt | 0.4851 | 0.3824 | 0.2015 | 0.5373 | -3.5680 |
| - *w/* example | 0.5187 | 0.4021 | 0.2258 | 0.5554 | -3.2842 |
| CoT Prompt | 0.5135 | 0.4077 | 0.2331 | 0.5484 | -3.2474 |
| - *w/* example | 0.5519 | 0.4354 | 0.2662 | 0.5897 | -2.7762 |
| *Ours* | **0.5814** | **0.4715** | **0.2933** | **0.6232** | **-2.3652** |
| *ESConv* | | | | | |
| Direct Prompt | 0.5279 | 0.4090 | 0.2386 | 0.5631 | -3.2760 |
| - *w/* example | 0.5632 | 0.4376 | 0.2658 | 0.5903 | -2.7149 |
| CoT Prompt | 0.5695 | 0.4437 | 0.2718 | 0.5997 | -2.6782 |
| - *w/* example | 0.6068 | 0.4912 | 0.3105 | 0.6431 | -2.1365 |
| *Ours* | **0.6324** | **0.5127** | **0.3315** | **0.6721** | **-1.8863** |

This suggests that the quadruple reward and tiered reward hierarchy implemented in the WeRG method enable LLMs to maximize the utility of high-quality data while compensating for the inadequacies of the substandard data during the fine-tuning process for ConvINT data generation.

### 4.4.2 HUMAN EVALUATION RESULTS

To complement automatic evaluation, we further conduct human evaluations on the generated ConvINT examples with three student annotators. For both the DuRecDial and ESConv datasets, we randomly sampled 50 conversations from their respective test sets for validation. The annotators were asked to rate the performance of various methods. The evaluation results are reported in Table 2, which intuitively reveals the following findings: (1) It is evident that our proposed method consistently outperforms the baseline methods across all three human evaluation metrics, affirming the efficacy and practicality of our approach in generating high-quality ConvINT data. (2) We find that the WeRG mechanism, by applying quadruple rewards that separately emphasize different aspects as formulated in the ConvINT framework, effectively captures comprehensive information within conversations, including emotional cues. This nuanced approach leads to notable improvements, particularly in emotional support conversations, where our method demonstrates the most significant performance enhancements. Overall, the human evaluation results are consistent with those of the automatic evaluations, demonstrating that our method adeptly fine-tunes LLMs to generate ConvINT data of superior quality.

Table 2: Human evaluation results for ConvINT generation. The scores, ranging from 0 to 5, represent averages across all samples rated by all annotators. $\mathcal{K}$ represents Fleiss' Kappa (Fleiss, 1971), indicating a fair to moderate level of inter-annotator agreement ($0.2 < \mathcal{K} < 0.6$).

| Methods | DuRecDial | | | ESConv | | |
|---|---|---|---|---|---|---|
| | Info. | Und. | Con. | Info. | Und. | Con. |
| Direct Prompt | 2.88 | 3.74 | 2.55 | 2.52 | 3.17 | 2.75 |
| w/ example | 3.26 | 3.93 | 2.72 | 2.79 | 3.33 | 3.03 |
| CoT+Prompt | 3.31 | 4.05 | 2.83 | 2.76 | 3.40 | 2.97 |
| w/ example | 3.45 | 4.24 | 2.95 | 2.92 | 3.58 | 3.26 |
| *Ours* | **3.71** | **4.38** | **3.62** | **3.55** | **4.06** | **3.78** |
| $\mathcal{K}$ | 0.47 | 0.42 | 0.45 | 0.39 | 0.49 | 0.42 |

### 4.5 IN-DEPTH ANALYSIS

### 4.5.1 ABLATION STUDIES

We conduct comprehensive ablation studies on the essential designs in our method—specifically, (1) the composition of weak supervision signals $\mathcal{D}_{\text{WeRG}}$ and (2) the reward module $r_c$—to analyze their individual contributions to overall generation performance using the DuRecDial dataset. The

Table 3: Ablation study results for ConvINT generation on the DuRecDial dataset. $w/o$ denotes the model fine-turned without the corresponding data source.

| Methods | F1 ↑ | BLEU1 ↑ | BLEU2 ↑ | BERTScore ↑ | BARTScore ↓ |
|---|---|---|---|---|---|
| *Ours* | **0.5814** | **0.4715** | **0.2933** | **0.6232** | **-2.3652** |
| - *w/o* $\mathcal{D}_{\text{coarse}}$ | 0.5744 | 0.4590 | 0.2811 | 0.6032 | -2.6276 |
| - *w/o* $\mathcal{D}_{\text{mid}}$ | 0.2355 | 0.1486 | 0.0832 | 0.2253 | -4.5094 |
| - *w/o* $\mathcal{D}_{\text{fine}}$ | 0.5488 | 0.4303 | 0.2622 | 0.5797 | -2.8361 |
| - *w/o* $r_c$ | 0.5347 | 0.4172 | 0.2430 | 0.5526 | -3.1249 |

experimental results are detailed in Table 3. In the first setting, we selectively remove three types of supervision signals ($\mathcal{D}_{\text{coarse}}$, $\mathcal{D}_{\text{mid}}$, and $\mathcal{D}_{\text{fine}}$) from the fine-tuning dataset, where $w/o$ denotes the configuration lacking the corresponding signals. As demonstrated in Table 3, excluding different sources of supervision from $\mathcal{D}_{\text{WeRG}}$ generally degrades the generation performance across both content-based and similarity-based evaluation metrics. In particular, the absence of supervision $\mathcal{D}_{\text{mid}}$, crucial for laying foundational insights into the ConvINT data, leaves the fine-tuning phase without essential guidance to extract the necessitated information that aligns with the defined ConvINT framework, leading to the most significant performance degradation. This suggests the effectiveness of these supervision signals with varying levels of annotated labels in supporting the model to generate higher-quality ConvINT data. In the second setting, we omit the quadruple reward $r_c$ with its differential reward hierarchy during the model fine-tuning, which results in a notable decrease in performance. We hypothesize this can be attributed to the lack of explicit signals that enable the model to discern between coarse-to-fine annotated data without the differential rewards.

### 4.5.2 EFFECT OF PROPORTION OF FINE-ANNOTATED DATA

In this section, we explore the effects of altering the proportion of human annotations $\mathcal{D}_{\text{fine}}$ on model performance. In the standard experimental setting, we include human annotations that comprise 10% of the total dataset (*i.e.*, $|\mathcal{D}_{\text{fine}}|/N = 10\%$), primarily due to the costs associated with human annotators. Considering the pivotal role this high-quality data plays in steering the fine-tuning process towards generating more comprehensive ConvINT data, we experimentally increase this ratio to further examine its impact on model training using the DuRecDial dataset. Table 3 illustrates the performance trends across various ratios of fine-annotated ConvINT data. Notably, as the proportion of $\mathcal{D}_{\text{fine}}$ increases, the model performance improves with sta-

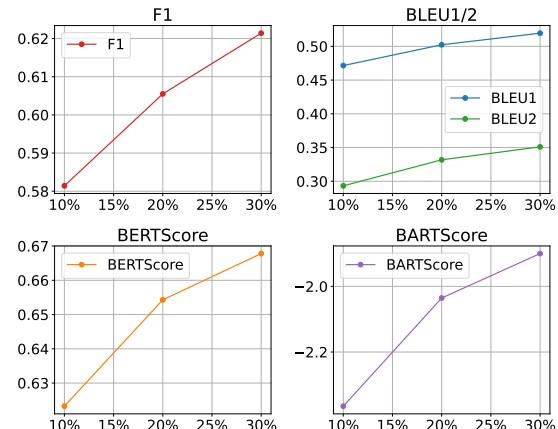

Figure 3: The impact of the proportion of fine-annotated data, ranging from 10% to 30%.

ble gains. This suggests that while the quantity of fine-annotated data $\mathcal{D}_{\text{fine}}$ is significantly less than $\mathcal{D}_{\text{coarse}}$ and $\mathcal{D}_{\text{mid}}$, it provides detailed insights into the human-preferred ConvINT data, continuously enhancing generation performance.

### 4.5.3 EFFECT OF CONVINT ON DOWNSTREAM APPLICATIONS

We further validate the effectiveness of applying the ConvINT data generated by the WeRG method to downstream conversational applications, specifically enhancing response generation in target-driven scenarios. We conduct experiments on the DuRecDial dataset by directly incorporating the ConvINT data into the inputs of the response generation model to enhance its output capabilities. Experimental results, detailing both dialogue-level and turn-level automatic evaluations, are presented in Table 4. By elucidating user utterances into fine-grained aspect information, our ConvINT framework markedly improves the ability of downstream response generation models, demonstrat-

Table 4: Automatic evaluation results for the downstream response generation task on the DuRecDial dataset, utilizing ChatGPT as the backbone generation model. ***CoT ConvINT*** denotes the CoT Prompt enhanced by the proposed ConvINT framework.

| Methods | F1 ↑ | BLEU1 ↑ | BLEU2 ↑ | SR ↑ | Avg. Truns ↓ |
|---|---|---|---|---|---|
| Direct Prompt | 0.4297 | 0.3716 | 0.2147 | 0.7686 | 4.97 |
| CoT Prompt | 0.4427 | 0.3815 | 0.2243 | 0.7952 | 3.86 |
| ***CoT ConvINT*** | **0.4785** | **0.4107** | **0.3187** | **0.8537** | **3.37** |

Table 5: Automatic evaluation results for the downstream response generation task on the ESConv dataset. $w/o$ indicates the removal of the corresponding fine-grained aspect from the ConvINT during integration into generating responses.

| Methods | F1 ↑ | BLEU1 ↑ | BLEU2 ↑ | SR ↑ | Avg. Truns ↓ |
|---|---|---|---|---|---|
| ***CoT ConvINT*** | **0.2979** | **0.2258** | **0.1370** | **0.8445** | **3.88** |
| - *w/o [SITUATION]* | 0.2904 | 0.2158 | 0.1265 | 0.8292 | 4.10 |
| - *w/o [EMOTION]* | 0.2284 | 0.1758 | 0.0865 | 0.7692 | 5.34 |
| - *w/o [ACTION]* | 0.2746 | 0.2090 | 0.1205 | 0.8023 | 4.45 |
| - *w/o [KNOWLEDGE]* | 0.2679 | 0.1988 | 0.1141 | 0.7923 | 4.25 |

ing the advantages of interpreting conversations in semi-structured natural language forms. Leveraging ConvINT, these models adeptly steer the flow of conversations by aligning subsequent turns with users' needs, thereby optimizing responses at each interaction to boost user engagement and successful target completion. Overall, the ConvINT framework lays a solid foundation for developing more sophisticated and effective conversational agents.

### 4.5.4 EFFECT OF DIFFERENT FINE-GRAINED ASPECTS IN THE CONVINT FRAME

The proposed ConvINT framework primarily establishes a multidimensional taxonomy, delving into aspects of situation, emotion, action, and knowledge to facilitate a comprehensive and multifaceted understanding of user utterances. To assess the individual contributions of these fine-grained aspects, we conduct experiments on the ESConv dataset by selectively omitting each of the four distinct aspects when applying the ConvINT framework to enhance downstream response generation. Results presented in Table 5 indicate a noticeable drop in performance whenever any detailed aspect is removed from the ConvINT framework. Notably, within the context of emotional support conversations, the removal of the ***[EMOTION]*** aspect—which is essential for revealing users' emotional cues throughout the conversation process—leads to the most substantial decrease in performance as the response generation model lacks specific guidance to tailor responses to users' emotional expectations. This underscores the potential of the ConvINT framework to support the customization of conversational agents for various real-world scenarios, aiding these agents in accurately grasping users' diverse needs and delivering effective responses.

## 5 CONCLUSION

In this work, we present a comprehensive exploration of conversational understanding by introducing the ConvINT framework, a novel fine-grained and aspect-aware formalism for understanding user intentions in intricate conversations. Building upon the ConvINT framework, we further devise a WeRG mechanism, which synergistically integrates diverse sources of coarse-to-fine ConvINT annotations as weak supervision signals. By assigning varying quadruple rewards to each data source, conditioned on the detail of the annotations, WeRG facilitates the generation of high-quality ConvINT data. Generally, our method not only advances the capabilities of conversational agents in dialogue understanding but also offers insights into effectively leveraging coarse-to-fine supervision signals for generating large-scale, high-quality data—a crucial step towards developing sophisticated conversational agents. Extensive experiments demonstrate the advantages of the ConvINT framework and confirm the superiority of the proposed WeRG approach.

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

## A   IMPLEMENTATION DETAILS

For the construction of the dataset $\mathcal{D}_{\text{WeRG}}$, we employ *gpt-3.5-turbo* as the mid annotator to generate $\mathcal{D}_{\text{mid}}$ in our experiments. To ensure deterministic outputs during the acquisition of ConvINT annotations, the temperature parameter is fixed at 0, and the output is limited to a maximum of 1000 tokens. All other parameters are kept at their default settings. The prompts are designed to guide the LLMs, as detailed in Appendix B. For the dataset $\mathcal{D}_{\text{fine}}$, we randomly sample 10% of the conversations from the original dataset for fine-grained annotations.

For ConvINT policy model training, we use *llama-2-7b* as the backbone model and apply LoRA fine-tuning. The model is fine-tuned for 3 epochs on the constructed dataset $\mathcal{D}_{\text{WeRG}}$ using the AdamW optimizer, with a learning rate initialized at $6.7 \times 10^{-5}$ and 100 warm-up steps. The fine-tuned parameters are saved every 1000 steps for subsequent evaluations. For the LoRA configuration, the rank is set to 8, the scaling factor to 16, and the dropout rate to 0.05. In the few-shot baseline setting, we utilize a one-shot demonstration randomly selected from the manually annotated dataset $\mathcal{D}_{\text{fine}}$.

For the reward setting, since the reward weight term in Equation (4) $\left(\exp\left(\frac{r_c}{\beta}\right)\right)$ remains constant within each class, we simplify the process by aligning the weights with the reward hierarchy $(r_{\text{fine}}) > r_{\text{mid}} > r_{\text{coarse}}$, assigning quadruple weights of $\langle 1.0, 1.0, 1.0, 1.0 \rangle$ to $\mathcal{D}_{\text{fine}}$, $\langle 0.5, 0.5, 0.5, 0.5 \rangle$ to $\mathcal{D}_{\text{mid}}$, and $\langle 0.1, 0.1, 0.1, 0.1 \rangle$ to $\mathcal{D}_{\text{coarse}}$ for the conversation recommendation scenario. For emotional support conversations, we emphasize the emotion aspect, assigning fine-grained aspect weights of $\langle 0.9, 1.0, 0.9, 0.9 \rangle$ to $\mathcal{D}_{\text{fine}}$, $\langle 0.4, 0.5, 0.4, 0.4 \rangle$ to $\mathcal{D}_{\text{mid}}$, and $\langle 0.05, 0.1, 0.05, 0.05 \rangle$ to $\mathcal{D}_{\text{coarse}}$.

## B   PROMPT DETAILS

The prompts utilized in our experiments are formulated as follows:

### B.1   DIRECT PROMPT

Please extract the conversational intentions based on the target-driven conversation provided below, where the {**target_goal**} guides the conversation. The intentions should concisely capture the user's focus conveyed in the [USER]-marked utterances. For each user utterance, identify the four aspects of user intentions—[SITUATION], [EMOTION], [ACTION], and [KNOWLEDGE]—and label them accordingly.

Please mark the input conversation according to the requirements and examples, ensuring each aspect is clearly addressed and provided. The marked intention numbers must strictly correspond one-to-one with conversation turns, with no merging or omissions allowed.

Example:

Conversation: ${**Conversation**}

ConvINT: ${**ConvINT**}

Input:

Conversation: ${**Conversation**}

ConvINT:

### B.2   COT PROMPT

Description: I want you to apply your expertise in philosophy, psychology, and cognitive science to analyze and extract user intentions from a target-driven conversation, where the AI aims to make a {**target_goal**} to the user. The conversation is target-driven, meaning it strategically shifts towards the AI's goal.

Requirements: User intentions should succinctly reflect the user's focus conveyed within [USER]-marked utterances during conversations. Below are the detailed definitions and marking requirements for four aspects of user intentions:

[SITUAION]: Describe any physical or situational context mentioned by the user. If not applicable, mark as [SITUATION]: None.

[EMOTION]: Capture any emotional states or evaluations expressed by the user. If no emotions are expressed, mark as [EMOTION]: None.

[ACTION]: List any actions the user mentions taking to achieve the goal. If no actions are taken, mark as [ACTION]: None.

[KNOWLEDGE]: Identify entities and relevant knowledge mentioned in the conversation. If no specific knowledge is referenced, mark as [KNOWLEDGE]: None.

Please mark the input conversation according to the requirements and examples, ensuring each aspect is clearly addressed and provided. The marked intention numbers must strictly correspond one-to-one with conversation turns, with no merging or omissions allowed.

Example:

Conversation: ${**Conversation**}

ConvINT: ${**ConvINT**}

Input:

Conversation: ${**Conversation**}

ConvINT:

### B.3   PROMPT TO RESPONSE GENERATION

Your task is to generate the next-turn response based on the provided ConvINT and Context. The Context is part of a **scenario** conversation between a user and an AI agent. The AI's goal is to guide the conversation toward achieving the {**target_goal**}. Generate a dialogue with frequent topic changes to extend the conversation over multiple rounds. Throughout the chat, smoothly switch topics back to the target goal, making the interaction feel like a natural, real-life conversation. Be specific in your responses. In the example below, [User] and [SYS] indicate the speaker of each turn.

Example:

Target: $**Target**

Conversation: ${**Conversation**}

ConvINT: ${**ConvINT**}

Input:

Target: $**Target**

Conversation: ${**Conversation**}

ConvINT: ${**ConvINT**}

Next-turn response:

## C   CASE STUDY

The following presents cases of existing intention interpretations and the generated ConvINT examples.

**Target Music:** Departing One

**Conversations:**

[SYS] Hello, Mr.Si! How are you doing?

[USR] 1. Hello! I'm good. It's just that the work is very boring.

[SYS] You can do something to enrich your life when you are bored.

[USR] 2. OK, I will. Thank you for reminding me!

[SYS] You can go to see a movie when you are bored. The movie 1:99 Shorts is wonderful. Give it a try!

[USR] 3. Not really for movie at the moment

[SYS] It's starred by Jacky Cheung who has won the Golden Horse Award for Best Actor.

[USR] 4. I like him very much.

[SYS] You've got good taste. He is a real singer.

[USR] 5. Thanks! The point is that he's very excellent. That's why I like him so much.

[SYS] Do you want to listen to Love You More Each Day by him? It's a golden melody in Hong Kong. It's very catchy.

[USR] 6. I've listened to it. Are there any other songs?

[SYS] Yes. How about Departing One? There are too many details. This brewer is gaudy but not pretentious.

[USR] 7. The name sounds so sad!

[SYS] You'll know after listening. Everyone feels differently.

[USR] 8. "OK, I'll give it a try.

**Existing Interpretations:**

1. **[Action]** Greetings **[Topic]** None

2. **[Action]** Greetings **[Topic]** None

3. **[Action]** Movie recommendation **[Topic]** 1:99 Shorts

4. **[Action]** Chat about stars **[Topic]** Jacky Cheung

5. **[Action]** Chat about stars **[Topic]** Jacky Cheung

6. **[Action]** Music recommendation **[Topic]** Love You More Each Day

7. **[Action]** Music recommendation **[Topic]** Departing One

8. **[Action]** Music recommendation **[Topic]** Departing One

**ConvINT:**

**Turn 1:**
**[Situation]** Boring work
**[Emotion]** Neutral, but implies dissatisfaction with work
**[Action]** Describes current emotional state regarding work
**[Knowledge]** None

**Turn 2:**
**[Situation]** None
**[Emotion]** Agreeable
**[Action]** Acknowledges the assistant's suggestion
**[Knowledge]** None

**Turn 3:**
**[Situation]** None
**[Emotion]** Disinterest in movies at the moment
**[Action]** Declines the movie suggestion
**[Knowledge]** The movie 1:99 Shorts

**Turn 4:**
**[Situation]** None
**[Emotion]** Positive, expresses admiration
**[Action]** Expresses liking for Jacky Cheung
**[Knowledge]** Jacky Cheung, Golden Horse Award for Best Actor

**Turn 5:**
**[Situation]** None
**[Emotion]** Positive, expresses strong admiration
**[Action]** Explains the reason for liking Jacky Cheung
**[Knowledge]** Jacky Cheung's excellence

**Turn 6:**
**[Situation]** None
**[Emotion]** None
**[Action]** Requests other song suggestions after acknowledging previous experience
**[Knowledge]** The song Love You More Each Day

**Turn 7:**
**[Situation]** None
**[Emotion]** Emotional anticipation, slight negativity due to the sad connotation of the song's name
**[Action]** Reacts to the name of the song
**[Knowledge]** The song Departing One

**Turn 8:**
**[Situation]** None
**[Emotion]** Willingness to try
**[Action]** Agrees to try listening to the suggested song
**[Knowledge]** None

