# OpenReview forum: "ConvINT: A Semi-Structured Intention Framework for Conversational Understanding"
_ICLR.cc/2025/Conference — ICLR 2025 Conference Withdrawn Submission_

### Official Review · Reviewer_ikwk · 2024-10-18

**Soundness:** 3
**Presentation:** 3
**Contribution:** 3
**Rating:** 6
**Confidence:** 4

**Summary:**

The paper presents a semi-structured framework, ConvINT, for intent recognition, introducing four key aspects inspired by psychological and cognitive intention theories. Additionally, the authors introduce WeRG, which scales ConvINT annotation while maintaining high quality.

**Strengths:**

- The paper proposes ConvINT, a more fine-grained framework for intent recognition
- The paper presents WeRG, which allows data synthesis from different source for model fine-tuning
- The paper is easy to follow

**Weaknesses:**

- Typo error l.350: "((Appendix B)", redundant "("
- l.367-l.369: from Table 1, I don't see that you differentiate the zero-shot CoT and few-shot CoT. (p.s. For evaluation, the deployed automated metrics are similarity-based or matching-based. Considering LLM-as-judge might be sensible here as well.)

**Questions:**

- I'm confused about the first formulation, especially the subscript. I assume each data point contain multiple dialogues, consisting of user utterances and responses. Then $x_{i, t}$ should be the t-th utterance from the i-th instance, while $y_{i, t}$ is the corresponding response?
- Is there a specific reason for choosing the DuRecDial dataset? Given that it is a dataset for conversational recommendations, I would imagine it might lack sufficient emotional or knowledge-based content, which are key aspects of your approach. Could you clarify why this dataset was selected? In addition, ESConv focuses on emotional support conversations, which should naturally contain more emotional information. Could you clarify why the informativeness score is lower compared to DuRecDial (Table 2)?
- l.321: How large is your test set? How many human annotators participated in the annotation process, and what was the level of inter-annotator agreement? Additionally, could you provide details about the annotation instructions?
- You specify the number of annotations in l.334-335. Could you clarify the background of the annotators? For the user study (Table 2), how many annotators evaluated each instance?
- In l. 445–446, you mention that the BARTScore of $w/o$ $D_{mid}$ and $w/o$ $r_c$ decreases (Table 3), which suggests improved performance. Could you provide clarification on this observation?
- In l. 448, I'm uncertain if it's fair to draw that conclusion, as the training data is dominated by $D_{mid}$, which should be the most influential factor affecting performance.

---

### Official Review · Reviewer_YTVM · 2024-10-28

**Soundness:** 2
**Presentation:** 3
**Contribution:** 2
**Rating:** 3
**Confidence:** 3

**Summary:**

This work studies conversational understanding (CU).

Unlike previous works that always use specified intent classes or slot-value pairs, this work uses LLMs to produce more flexible natural language CU understanding texts.

Next, this work is inspired by psychological and cognitive intention theories (Schroder et al., 2014) and proposes four CU aspects. Then, this work proposes a weakly supervised RL fine-tuning method to use three sources of training data: (coarse mapping, LLM annotation, and human annotation ).

Finally, experiments are conducted on two datasets.

**Strengths:**

1. This work is well-organized and easy to follow.
2. This work draws from interdisciplinary intention theories to formulate its task.  Subsequently, four CU aspects are identified.
3. ConvINT/WeRG can simultaneously use three sources of training data.
4. Many analyses about the sources of training data

**Weaknesses:**

1. The novelty is limited. The contribution can be summarized as 1) a new 4-aspect CU template and 2) combining three sources of training data.
2. In the experiments, it seems like the COT baseline prompt lacks the process of COT. It is more like an In-Context Learning prompt.
3. The experimental setting is unfair. Two baselines are non-trained, prompt-based methods, but the proposed method has included a training process with supervised data. This means this work lacks settings of other (weakly-)supervised baselines.
4. In the Table 3 Ablation Study, we can find the LLM-annotated $D_{mid}$ can heavily impact the result. This can also indirectly prove that LLMs perform well in CU tasks with a well-defined prompt. However, the baseline setting does not include such a well-defined prompt design. This means this work lacks settings of other necessary baseline prompts.
5. The experiment lacks enough discussion about their training methodology. Most analyses are related to the sources o

**Questions:**

minor issues:

1.  Table 1: BARTScore is also higher is better.

---

### Official Review · Reviewer_FWex · 2024-11-02

**Soundness:** 2
**Presentation:** 2
**Contribution:** 2
**Rating:** 3
**Confidence:** 4

**Summary:**

This work proposes ConvINT, a semi-structure intention framework that offers fine-grained understanding of user intentions. Concretely, it proposes to categorize user intention into four categories: situation, emotion, action and knowledge. Building upon this rich representation, this work introduces a weakly supervised reinforced generation method that scales the ConvINT annotation. Experimental results show that the proposed method improved LLMs’ ability to understand user intentions on two datasets.

**Strengths:**

* This work is grounded in psychological and cognitive intention theories and produce a fine-grained ConvINT to represent user intentions.
* The proposed WeRG method takes into account coarse-to-fine labels and introduced three different levels to capture fine-level, mid-level and coarse-level labels. Such design holds the potential to balance different types of data quality and annotation cost.
* Detailed ablation studies are provided to demonstrate the effectiveness of each component, such as the use of prompting and the necessity of different levels of labels in ConvINT generation, as well as the amount of fine-annotated data.

**Weaknesses:**

* WeRG relies on three levels of labels and their relative quality difference to learn this hierarchical information with regard to rewards. However, it is unclear how such a mixture of three different levels of labels should be, and how model training is sensitive to such data mixture.
* One of the justifications for using coarse and mid level labels is for cost-effective considerations. However, current experimental design didn’t provide much comparison of cost and effectiveness with regard to different design choices here.
* The current baselines only include vanilla prompting and chain of thought prompting. I wonder why no other training-based methods are not used as baselines. It seems WeRG can also be viewed as synthetic data generated from weaker models; would any prior synthetic data generation methods or distant supervision be good alternatives here?

**Questions:**

* Many components of the current paper require further justification or motivation. For instance, it is a bit hard to follow the training details around Figure 2. Similarly,  ConvINT framework is described as “According to the intention theories in psychological and cognitive sciences”, but it is unclear how such grounding or design works that lead to the current triplets of situation, emotion, action and knowledge. Why is this detailed breakdown essential and can provide add-on value for conversational understanding?

* How coarse-, mid-, and fine- level labels are derived and evaluated is missing key details. What’s the quality of these mid-level labels? Are they noisy and to what extent? For fine-level labels, who are the annotators, how are annotators compensated, and what are the annotation agreements among these annotators?

---

### Official Review · Reviewer_eZnx · 2024-11-03

**Soundness:** 3
**Presentation:** 2
**Contribution:** 3
**Rating:** 5
**Confidence:** 4

**Summary:**

This paper introduces ConvINT (Conversational INTention), a semi-structured framework for enhancing intention recognition in conversational AI. ConvINT employs a fine-grained, aspect-aware analysis of user input across four key dimensions: situation, emotion, action, and knowledge. To efficiently expand ConvINT annotations in large datasets, the authors develop the WeRG (Weakly Supervised Reinforcement Generation) approach, which synergizes multiple annotation sources—including hard mapping, LLMs, and human annotations—for fine-tuning the model. Experimental results demonstrate that the ConvINT framework improves conversational understanding and confirms the effectiveness of the WeRG method.

**Strengths:**

- The ConvINT framework effectively incorporates cognitive and psychological intention theories, making it suitable for handling emotional and nuanced interactions in real-world conversational AI.

- The WeRG method for ConvINT annotations, which combines weak supervision from multiple sources with hierarchical rewards, ensures high-quality annotations while reducing dependency on costly human resources.

**Weaknesses:**

- Comparison Ambiguity: It seems that Direct Prompt and CoT Prompt were applied with llama2-7b (please clarify if this is incorrect). The comparative value of these methods against the WeRG training method remains unclear. Additionally, as GPT-3.5-turbo was used for annotation, the lack of detail on the quality of this data raises questions about the reliability of D_{mid}.

- Limited Analysis: The ablation study on data sources and the experiments for downstream applications are conducted exclusively on DuRecDial, while the ablation study on aspects is performed only on ESConv. However, the noticeable drop when ‘[Emotion]’ is removed from ESConv seems expected and does not provide meaningful insight. Furthermore, there is no comparison with traditional interpretation methods in the downstream experiments, nor is there any analysis of key failure cases, such as poorly generated ConvINTs or unsatisfactory responses based on accurate ConvINTs, which could yield valuable insights.

- Lack of Experimental Detail:  Specific details of the ConvINT labels for the test set and the evaluation by the three student annotators are missing. Additionally, it is unclear how the Success Rate (SR) metric is calculated.

**Questions:**

1. When constructing the ConvINT data, what is the quality of the content generated by GPT-3.5 as an annotator for each aspect? Additionally, how was the human annotation process conducted?

2. Why was the ablation study on data sources and downstream applications limited to DuRecDial, and why was the aspect-based ablation study conducted solely on ESConv? Could expanding these studies across both datasets provide more comprehensive insights?

3. In the downstream experiments, is there any comparison with traditional interpretation methods or analysis of key failure cases, such as poorly generated ConvINTs or responses that fall short despite accurate ConvINTs? If not, would these additions offer further insights?

4. What are the specific details of ConvINT labels for the test set and the evaluation conducted with the three student annotators? Moreover, how is the Success Rate (SR) calculated?

5. Line 293: KL regularised RL -> KL-regularized RL

6. Line 350: one bracket

---

### Note · Authors · 2024-11-15

I have read and agree with the venue's withdrawal policy on behalf of myself and my co-authors.